Validation and description of two new north-western Australian Rainbow skinks with multispecies coalescent methods and morphology

Afonso Silva Ana C. anacatarina.as@gmail.com 1 2
Santos Natali 3
Ogilvie Huw A. 1 4
Moritz Craig 1
1 Division of Ecology and Evolution, Research School of Biology and Centre for Biodiversity Analysis, Australian National University , Acton , ACT , Australia
2 cE3c—Centre for Ecology, Evolution and Environmental Changes, Faculdade de Ciências, Universidade de Lisboa , Lisboa , Portugal
3 Universidade Federal do ABC , Santo André , SP , Brazil
4 Centre for Computational Evolution, University of Auckland , Auckland , New Zealand
Prentis Peter
Electronic publication date: 2017 Aug 29
Publication date: 2017
Volume: 5
Electronic Location ID: e3724
Received 2017 Feb 8; Accepted 2017 Aug 1
Copyright: ©2017 Afonso Silva et al.
Copyright year: 2017
Copyright holder: Afonso Silva et al.
License: This is an open access article distributed under the terms of the Creative Commons Attribution License, which permits unrestricted use, distribution, reproduction and adaptation in any medium and for any purpose provided that it is properly attributed. For attribution, the original author(s), title, publication source (PeerJ) and either DOI or URL of the article must be cited.
License URL: https://creativecommons.org/licenses/by/4.0/

Keywords: Integrative taxonomy, Carlia, Species delimitation, Australia, Multispecies coalescent, BFD* StarBeast2

Funding: Australian Biological Resources Study Australian Research Council ARC FL110100104 FCT SFRH/BD/88740/2012 This research was supported by grants from the Australian Biological Resources Study to CM and Scott Keogh, and from the Australian Research Council to CM (ARC FL110100104). ACAS is supported by the FCT grant SFRH/BD/88740/2012. The funders had no role in study design, data collection and analysis, decision to publish, or preparation of the manuscript.

==============================
While methods for genetic species delimitation have noticeably improved in the last decade, this remains a work in progress. Ideally, model based approaches should be applied and considered jointly with other lines of evidence, primarily morphology and geography, in an integrative taxonomy framework. Deep phylogeographic divergences have been reported for several species of Carlia skinks, but only for some eastern taxa have species boundaries been formally tested. The present study does this and revises the taxonomy for two species from northern Australia, Carlia johnstonei and C. triacantha. We introduce an approach that is based on the recently published method StarBEAST2, which uses multilocus data to explore the support for alternative species delimitation hypotheses using Bayes Factors (BFD). We apply this method, jointly with two other multispecies coalescent methods, using an extensive (from 2,163 exons) data set along with measures of 11 morphological characters. We use this integrated approach to evaluate two new candidate species previously revealed in phylogeographic analyses of rainbow skinks (genus Carlia) in Western Australia. The results based on BFD StarBEAST2, BFD* SNAPP and BPP genetic delimitation, together with morphology, support each of the four recently identified Carlia lineages as separate species. The BFD StarBEAST2 approach yielded results highly congruent with those from BFD* SNAPP and BPP. This supports use of the robust multilocus multispecies coalescent StarBEAST2 method for species delimitation, which does not require a priori resolved species or gene trees. Compared to the situation in C. triacantha, morphological divergence was greater between the two lineages within Kimberley endemic C. johnstonei, which also had deeper divergent histories. This congruence supports recognition of two species within C. johnstonei. Nevertheless, the combined evidence also supports recognition of two taxa within the more widespread C. triacantha. With this work, we describe two new species, Carlia insularis sp. nov and Carlia isostriacantha sp. nov. in the northwest of Australia. This contributes to increasing recognition that this region of tropical Australia has a rich and unique fauna.

Introduction

Cryptic species—when two or more distinct species are inaccurately classified under one species name (Bickford et al., 2007)—present great challenges for taxonomy and species delimitation due to the desirability of validating candidate species using multiple lines of evidence (Fujita et al., 2012). But for biodiversity assessment and conservation reasons the need to properly describe species diversity is greater than ever (Bickford et al., 2007). In the same way, there is a concern that molecular data may promote taxonomic inflation by ‘over splitting’ divergent populations into candidate species (Isaac, Mallet & Mace, 2004; Hedin, Carlson & Coyle, 2015). The creation of more reliable and robust species delimitation approaches in the last decade has attempted to address this concern (Rannala, 2015).

To more robustly infer species boundaries, the use of integrative taxonomy is increasingly common (Bickford et al., 2007; Padial et al., 2010). The objective of this approach is to corroborate taxonomic validity with independent, distinct types of evidence. Given deep genetic divergence, fixed morphological differences are not necessary to diagnose species boundaries since speciation itself does not require phenotypic characters to evolve at the same rate as the genome (Leaché & Fujita, 2010). Therefore, in taxa with inherently conservative morphology, it may be that the primary evidence for distinct species will come from genetic data.

Species delimitation consists of two potentially complementary approaches: discovery methods that do not require a priori assignment of samples before analysis, and validation methods that test hypotheses based on samples already assigned to candidate species (Ence & Carstens, 2011). When candidate lineages are already identified, validation approaches are more robust because they explicitly model the process of lineage diversification (Carstens et al., 2013). This is especially so when there is a substantial number of informative genes, independent of those used to suggest candidate taxa. Model-based multilocus approaches that use the multispecies coalescent (MSC) are advantageous because they account for coalescent processes when estimating phylogenetic relationships (Edwards et al., 2016). And for species delimitation, objective and transparent model-based approaches are relevant, because they have the potential to reduce investigator-driven biases (Fujita et al., 2012). These methods can consider gene tree incongruence due to incomplete lineage sorting, variation in molecular sequences and variation in demographic parameters (Leaché & Fujita, 2010). With this in mind, Carstens et al. (2013) recommend the best approach for species delimitation is to use multiple methods. Further, Rannala (2015) suggests that this should only be done when methods have algorithmically similar assumptions. However, we also note that MSC methods can over split—revealing high structured populations (or ephemeral species; Rosenblum et al., 2012)—rather than long isolated species, depending on the nature of the speciation process (Sukumaran & Knowles, 2017). Hence, species delimitation will always be more secure when taxa delimited using genetic methods are somehow corroborated by alternative sources of data (Oliver et al., 2014).

Previous work by Afonso Silva et al. (2017), which focused on understanding how phylogeographic structure and history differs between a climatic generalist and specialist, found two deeply divergent lineages within each of Carlia johnstonei Storr, 1974 and C. triacantha Mitchell, 1953 (Fig. 1). These sister taxa (Dolman & Hugall, 2008) have contrasting distributions, with the former being endemic to the Kimberley and the latter being widespread across northern Australia. The lineages within C. johnstonei are likely allopatric, with the nominal lineage (Johnstonei A) being found across the north and western Kimberley and the newly identified lineage (Johnstonei B) being endemic to islands off the coast of the northwest Kimberley (Fig. 1). Conversely, the two lineages of C. triacantha likely overlap geographically, with the nominal lineage being widespread across north and central Australia (Triacantha A) and the newly identified lineage (Triacantha B) found within the Kimberley and scattered locations in the central Northern Territory (Fig. 1).

Figure 1 Distribution map with used genetic samples and measured specimens for C. johnstonei (A) and for C. triacantha (B) lineages, and lineages relationships (C) as in Afonso Silva et al. (2017).

Triangles correspond to the genetic samples used in this study while circles correspond to specimens measured. Blue Johnstonei A, yellow Johnstonei B, green Triacantha A and purple Triacantha B. Tree obtained with 20 representative samples in ASTRAL and respective lineage bootstrap.

Species of Carlia from the Australian tropics generally have deep phylogeographic structure for both mtDNA and large numbers of exons (e.g., Potter et al., 2016), and, where contact zones have been examined in detail, there is evidence of strong reproductive isolation between the more deeply divergent (but phenotypically cryptic) lineages (Phillips, Baird & Moritz, 2004; Singhal & Moritz, 2013). However, recent species delimitation and taxonomic revisions have focussed more on Carlia from the eastern woodlands and placed a greater emphasis on morphology (e.g., Hoskin & Couper, 2012; Hoskin, 2014). There is a need to re-examine the systematics of northern Australian Carlia, and here we have the opportunity to exploit large multilocus datasets (Bragg et al., 2015) to do that integrated with morphology. This is particularly relevant for Kimberley biodiversity, since there have been recent efforts to discover and describe new species (Köhler, 2011; Oliver et al., 2014; Oliver et al., 2016; Andersen et al., 2014; Ellis, 2016) in this still relatively unknown and remote region in the northwestern of Australia.

We reanalyse the extensive multilocus data used in Afonso Silva et al. (2017) using robust species delimitation methods, together with morphological analysis to validate species hypotheses. Following Rannala (2015), we use three algorithmically similar methods to validate potential new species. We apply BPP (Yang & Rannala, 2014) and two approaches using Bayes Factors to test species hypotheses: a SNP based approach, BFD* (Leaché et al., 2014) using SNAPP (Bryant et al., 2012) and a sequence-based approach, BFD with the recently developed StarBEAST2 method (Ogilvie, Bouckaert & Drummond, 2017). We consider three potential species hypotheses: (i) only the two currently defined species are separated; (ii) a three-species hypothesis—two species corresponding to the two more deeply divergent lineages of C. johnstonei, but collapsing the less divergent lineages within C. triacantha; and (iii) a four-species hypothesis—all four lineages correspond to different species. Using an integrative taxonomic approach, we present and analyse morphological data to test for congruent differences between all identified genetic lineages. Considering all lines of evidence, we then formally describe the new species and identify diagnostic traits, for both morphology and gene sequences. Genetic diagnostic traits include SNPs from available mtDNA ND4 gene sequences (Afonso Silva et al., 2017), following Renner’s (2016) suggestion to provide simple genetic diagnostics, particularly for morphologically similar species groups.

Materials & Methods

We used exon capture data to perform validation analyses and sequences of the mtDNA ND4 gene to identify diagnostic SNPs, and, also measured, analysed and identified diagnostic morphological traits. We obtained sequences for the genetic data from Afonso Silva et al. (2017) (Dryad Digital Repository: http://datadryad.org/resource/doi:10.5061/dryad.jj1tt). These included mtDNA sequence data of 101 C. johnstonei and 99 C. triacantha throughout both species’ distribution, for which we had specimens to do morphological analysis (Table S1, Fig. S1).

See Afonso Silva et al. (2017) for more detail about how the exon capture data was obtained. In summary, the data was retrieved from a custom set of loci designed from transcriptomes of Carlia and a couple of related genera (Bragg et al., 2015). After similar processing to Bragg et al. (2015), the final dataset contained a total of 51 samples with average of 40× coverage and approximately 2,800 loci per sample. For the validation analyses, we retrieved data from the 20 geographically dispersed samples as used for species tree estimation in Afonso Silva et al. (2017) (Fig. 1, Table S1). These correspond to five individuals for each of the four lineages previously identified in Afonso Silva et al. (2017) (Fig. 1, Table 1), using the same C. amax samples as an outgroup (from Potter et al., 2016) that were used in that study. For these analyses, we required aligned haplotype sequences, for which we employed GATK (v3.3, McKenna et al., 2010) which was also used to identify heterozygous sites and mask sites with a low-quality genotype call (GQ < 20). Here, we generated phased haplotypes using the individual overlapping sequencing reads to phase heterozygous sites within target loci and then used one haplotype per sample in later analyses.

Table 1 Species delimitation support.

For BPP support is in posterior probabilities while for BFD StarBeast2 and BFD* SNAPP is based in Bayes Factors calculated using the two species model as the null model (two species support by comparing with the four species model).

	BPP gene set1	BPP v gene set2	StarBeast2 gene set1	StarBeast2 gene set2	SNAPP SNP set1	SNAPP SNP set2	
Two species	0	0	−318.10	−274.04	−4517.80	−4443.35	
Three species	0	0	223.35	203.79	3526.41	3370.49	
Four species	1	1	318.10	274.04	4517.80	4443.35	

We then used the EAPhy pipeline (Blom, 2015 v.1.2; https://github.com/MozesBlom/EAPhy) to realign, filter and export alignments with complete data into NEXUS and PHYLIP format, as well as two sets of SNPs in FASTA format (using 0.2 as maximum proportion of Ns for each site, one SNP chosen randomly per gene and excluding singletons).

Genetic species validation

We applied three multispecies coalescent validation approaches to investigate species boundaries: Bayesian Phylogenetics and Phylogeography (BPP v3.3; Yang & Rannala, 2014), BFD (Bayes factor delimitation; Grummer, Bryson & Reeder, 2014) StarBEAST2 using multilocus data (Ogilvie, Bouckaert & Drummond, 2017), and BFD* SNAPP using SNP data (Leaché et al., 2014).

For the BPP analysis, we randomly selected two exon sets (to avoid unforeseen biases), each with 100 loci of between 250 bp and 1,000 bp, to check for consistent results. The MSC assumes no recombination within loci, and free recombination among loci (Degnan et al., 2009). We are confident of satisfying the latter condition, as our exons are all derived from different genes (Bragg et al., 2015). Lanier & Knowles (2012) showed that intra-locus recombination had little effect in species-tree estimates under the MSC; however Potter et al. (2016) found that it can affect species delimitation. Hence, to further evaluate this effect, we used the program IMgc (Woerner, Cox & Hammer, 2007) to extract optimal recombination-filtered blocks (no four-gamete violations) and repeated BPP analysis for comparison. We performed joint Bayesian species delimitation and species tree estimation (method A11, Yang, 2015). This method uses the multispecies coalescent model to compare different models of species delimitation and species phylogeny in a Bayesian framework, accounting for incomplete lineage sorting due to ancestral polymorphism and gene tree species tree conflicts (Yang & Rannala, 2010; Yang & Rannala, 2014; Rannala & Yang, 2013). Ancestral population size parameters (theta) were set to gamma prior G(2, 1,000), with mean 2∕1,000 = 0.002 and the divergence time at the root of the species tree (tau) was assigned to G(2, 2,000), while the other divergence time parameters were assigned to the Dirichlet prior (Yang & Rannala, 2010: equation 2). Preliminary analyses run using different combination of gamma priors, as suggested in Yang (2015), produced similar results, suggesting that our results are robust to the priors used. The phylogeny obtained in Afonso Silva et al. (2017) was used as a starting tree and all columns in the alignment were used in the likelihood calculation. Each exon set analysis was independently run twice to confirm consistency between runs, with a burn-in of 50,000 and a sampling frequency of five iterations for a total of 500,000 generations.

Bayes factor delimitation (BFD; Grummer, Bryson & Reeder, 2014) is an approach that compares the marginal likelihoods of competing species delimitation hypotheses using Bayes factors. To apply this approach, we ran two MSC methods to test our three potential hypotheses using C. amax as an outgroup: (i) a scenario with two species (C. johnstonei and C. triacantha), (ii) a scenario with three species (lineages Johnstonei A, Johnstonei B and C. triacantha) and (iii) a scenario with four species (with both lineages from C. triacantha and C. johnstonei as separate species).

StarBEAST2 v0.13.5 is a recently released sequence-based approach that reconstructs species trees with more flexibility than BPP (Ogilvie, Bouckaert & Drummond, 2017), and so provides an alternative MSC method to investigate species delimitation with Bayes factors (BFD). To verify consistency, we randomly selected another two sets of exons, each with 20 loci between 250 and 1,000 bp. We then used jModelTest v2.1.10 (Guindon, Gascuel & Rannala, 2003; Darriba et al., 2012) to calculate nucleotide substitution model likelihood scores for each locus and to estimate optimal model using BIC (Supplemental Table S2). All BFD StarBEAST2 analyses were performed using a strict clock model, for 100,000,000 generations, with data sampled every 10,000 generations, the first 10% of each run was discarded as burn-in and priors as in Table S3. For each analysis, two BFD StarBEAST2 replicates were conducted to ensure convergence and assessed using ESS values with Tracer v1.6 (Rambaut et al., 2015). We used stepping-stone sampling (Leaché et al., 2014) to determine the marginal likelihoods of four, three and two species (plus outgroup). All stepping-stone analyses used 16 steps with a beta distribution α parameter of 0.1 to optimise the power posterior discretization (Xie et al., 2010). The resulting marginal likelihoods were then used to compute Bayes factors (Kass & Raftery, 1995), quantifying the support for each species delimitation hypothesis against all others under consideration. The final tree was obtained by combining posterior replicates with LogCombiner (Drummond & Rambaut, 2007) and summarised using maximum clade credibility trees, after exclusion of 10% burn-in, with TreeAnnotator v1.7.2 (Drummond & Rambaut, 2007).

To use an approach that considers evidence from all available loci, we selected two independent SNP sets by sampling one SNP at random from each locus out of 2,163 total available loci and estimated species trees for each scenario using SNAPP (Bryant et al., 2012). We ran all analysis for 500,000 generations sampling every 500, with two replicates to ensure convergence and priors as in Table S3. After assessing convergence between runs and exon sets we proceeded to Bayes factor delimitation as described previously.

Morphological data collection

We analysed 200 specimens from the Museum and Art Gallery of the Northern Territory (MAGNT), Museum Victoria (MV), South Australian Museum (SAM), Western Australian Museum (WAM) and recently-collected specimens held at the Australian National University (with ANU ethical approval number A2012/14) (Fig. 1, Table S1). All analysed specimens were also sequenced for the mtDNA ND4 gene in Afonso Silva et al. (2017) (Fig. S1), with a total of 66 examined specimens for Johnstonei A, 35 for Johnstonei B, 31 for Triacantha A and 68 for Triacantha B.

We examined five morphometric characters taken to the nearest 0.1 mm with Mitutoyo electronic callipers: snout-vent length (SVL), axilla-groin length (AGL), head length (HL) measured from anterior edge of tympanum to snout, head width (HW) measured at widest point of the head, and head depth (HD) measured at parietal scales. In order to minimize error, we used a dissecting microscope Leica MZ8 (equipped with camera Leica MC120 HD) for which forelimb (FLL) and hindlimb length (HLL) were measured through photographs using ImageJ (Abràmoff, Magalhães & Ram, 2004) (as in Fig. S3); as well as four additional smaller features: nasals separation (NS), ear aperture length (EAL), palpebral disc length (PDL) and eye to ear distance (EED) (as explained in Fig. S3).

We also assessed seven meristic characters using photographs: supralabials, infralabials, supraciliaries, lamellae under the 4th toe (from claw sheath to junction of 3rd and 4th toes), lamellae under the 3rd finger (from claw sheath to the junction of the 2nd and 3rd fingers), the mode of number of keels across the mid-dorsal line scales and the ear lobule numbers. These traits were counted as suggested by Cogger (2014) and similarly to Hoskin & Couper (2012). Measurements and scales were generally analysed from the left side of the specimen, unless prevented by damage or poor preservation. All described measurements were collected in millimetres (mm).

For the ensuing species descriptions, we also measured the tail length and the distance between prefrontals if not in contact, but these traits were not used in the morphological analysis due to high level of missing data. For the designated holotypes, we additionally counted the number of midbody scale rows, vertebral (from the occiput to the edge of the hind limb along the mid dorsal line) and ventral scales (from mental scale to the edge of cloaca).

Morphological analyses

We investigated the relationship of each linear measurement with size (per mtDNA lineage), by plotting each variable against SVL and by comparing box plots of raw and size corrected measurements. After removing samples with missing data, all measurements were log-transformed to reduce their variance allowing a more conservative assessment of differences between mtDNA lineages. We then extracted size-corrected residuals from regressions between SVL and each measurement as a size-corrected log-transformed dataset. We investigated normality and heteroscedasticity after variable correction using density plots, Shapiro–Wilk test and Levene’s test. Multivariate normality was assessed with the Henze-Zirkler’s Multivariate Normality Test in the MVN package (Korkmaz, Goksuluk & Zararsiz, 2014).

In order to assess the morphometric distinctiveness of these lineages, we conducted Principal Component analyses on the log-transformed and on the size-corrected log-transformed (excluding SVL) datasets for each species. We used the prcomp function (stats package) with all measurement variables centred and plotted principal component 1 (PC1) against PC2, with a 75% confidence ellipse probability threshold (ggplot2 package, Wickham, 2016).

To statistically evaluate whether the lineages are significantly different and which variables are contributing to this, we analysed log-transformed and size-corrected log-transformed measurements with a MANOVA, and confirmed the significance of non-normal variables with the non-parametric Wilcoxon test (stats package). Relevant meristic data was analysed independently with a generalized linear modelling with a Poisson distribution (stats package) since these are count data and not continuous variables.

Using the statistically significant measurement variables from the MANOVA, we tested the accuracy in predicting assignment of lineage by applying a linear discriminant analysis (LDA) with jackknife cross-validation implemented in the package MASS (Venables & Ripley, 2002). Due to the presence of non-normal variables, we also applied a Random Forest (RF) analysis using the package randomForest (Liaw & Wiener, 2002).

We investigated the effect of possible outliers in the data by calculating, for each of the variables, interquartile range scores (function scores in outliers package, Komsta, 2011) to identify samples with outliers and then perform a MANOVA with this dataset. Removing outliers decreases 14% and 6% of analysed specimens for C. johnstonei and C. triacantha, respectively. Since some of these outliers could represent expected phenotypic variation across these species distribution and the overall results were similar, we present the analyses with all individuals.

To account for the insufficient information on sex, we performed a linear model containing sex and mtDNA lineage, using the available sexed individuals, which showed no difference in SVL between males and females in either C. johnstonei or C. triacantha. This suggests sex differences cannot explain our observed results, so we also present the analyses with all individuals.

We performed all analyses in R v.3.3.1 (R Core Team, 2016) and all the data, input files, code and morphological results are available at https://dx.doi.org/10.6084/m9.Figshare.4621963.

Molecular diagnostics

Following the recommendation of Renner (2016), we visually identified diagnostic SNPs within the ND4 mtDNA gene using all Afonso Silva et al. (2017) sequences with Genbank accessions codes MF083173 –MF083508 in Geneious v.7.1.9 (http://www.geneious.com, Kearse et al., 2012). Using as a reference an available skink mitogenome from Scincella vandenburghi (Park et al., 2016), we selected the available diagnostic SNPs per lineage within each species, where the nucleotide difference would correspond to an amino acid substitution.

The electronic version of this article in Portable Document Format (PDF) will represent a published work according to the International Commission on Zoological Nomenclature (ICZN), and hence the new names contained in the electronic version are effectively published under that Code from the electronic edition alone. This published work and the nomenclatural acts it contains have been registered in ZooBank, the online registration system for the ICZN. The ZooBank LSIDs (Life Science Identifiers) can be resolved and the associated information viewed through any standard web browser by appending the LSID to the prefix http://zoobank.org/. The LSID for this publication is: urn:lsid:zoobank.org:pub:A7B29F16-079F-48BA-B4BE-3EC9A3D80D34. The online version of this work is archived and available from the following digital repositories: PeerJ, PubMed Central and CLOCKSS.

Figure 2 Species tree with topology from BFD StarBeast2 gene set1 presenting node posterior probabilities for the two sets of data used for all three MSC methods.

Results

MSC Species delimitation

All three MSC approaches assigned more support to the four-species hypothesis than either the two- or three-species hypotheses (Table 1).

Both BPP analyses, each with independent drawn sets of genes, yielded the same species tree (Fig. 2) and a posterior probability (PP) equal to 1 for five delimited species (all four lineages plus the outgroup). The analyses processed with IMgc to exclude blocks with no four-gamete violations from within alignments, returned similar results with PP = 1 for four lineages plus outgroup. However, while topology for the original datasets was as expected by 99% of the models (Fig. 2), for each gene set without recombining blocks only 64% and 85% of the models supported the same topology.

For both BFD StarBEAST2 and BFD* SNAPP, Bayes Factors (BF) were obtained by subtracting the two-species hypothesis from both the three-species hypothesis as well the four-species hypothesis, and multiplying the difference of marginal likelihoods by a factor of two.

The BFs for both the BFD StarBEAST2 and BFD* SNAPP analyses were >10 for the four-species hypothesis relative to the two- or three-species hypotheses (Table 1), which corresponds to decisive evidence for this model (Kass & Raftery, 1995). The marginal likelihood results were of similar magnitude across the two gene datasets for BFD StarBeast2 and across the two SNP datasets for BFD* SNAPP (Table 1, Fig. S2), although BFs were much higher for the latter.

The species tree topology with the main lineages was assessed in Afonso Silva et al. (2017) using the ASTRAL summary species tree method (Fig. 1), but here species trees were also estimated by BPP, StarBEAST2 and SNAPP. StarBEAST2 and SNAPP all returned majority support for the ASTRAL topology. For gene sets 1 and 2, StarBEAST2 support for the ASTRAL topology was 97% and 63%, respectively. Support was higher using SNAPP at 95% and >99% for SNP set 1 and 2, respectively.

Figure 3 PCA with log transformed (A, B) and size corrected (C, D) morphological measurements for C. johnstonei and for C. triacantha with colours by mtDNA lineage.

Morphological analysis

The morphological measurements suggest that snout-vent length (SVL) is an important differentiating trait between candidate species within each of C. johnstonei and C. triacantha (Fig. 3, Fig. S4). Thus, further analyses were conducted also using size-corrected log-transformed variables (Fig. S5), so we could assess if the lineages were statistically different after accounting for SVL differences. For multivariate analyses, individuals with missing data were removed and after size correction some variables were still not normal (Tables S4, S5), but were multivariate normal for both C. johnstonei (log-transformed HZ p-value = 0.056, size corrected HZ p-value = 0.121) and C. triacantha (log-transformed HZ p-value = 0.104, size corrected HZ p-value = 0.272).

In the PCA results for C. johnstonei with only log-tranformed data (including SVL), the first axis (PC1) explained 74.4% of the total variation with all variables loading uniformly (and hence size-related) and the second axis (PC2) only explained 7.2% of variation (Fig. 3A, Fig. S6A). By contrast, in the PCA with the size corrected dataset (and excluding SVL), PC1 explains 26.2% and PC2 16.8% of the variation (Fig. 3C, Fig. S6C). The log-transformed PCA shows more evidence of clustering by lineage than does the size-corrected PCA. Together these observations point to a high similarity in shape, relative to divergence in body size. For C. triacantha, similar results were obtained (Fig. 3B, Fig. S6B). The proportions of variance explained for log transformed analysis were PC1 = 74.3% and PC2 = 7.2%; whereas, for the size corrected analysis, PC1 = 19.5% and PC2 = 15.4% (Fig. 3D, Fig. S6D).

Using MANOVA, we assessed whether morphological measurements differences between lineages were significant (for more detail see Tables S4–S5). For both species, the MANOVA confirmed that size (logSVL) differs between lineages in each species (p = 1.05 × 10−6 in C. johnstonei; p = 6.96 × 10−3 in C. triacantha). For size-corrected data, head depth (p = 1.36 × 10−3), nasal separation (p = 9.02 × 10−3), forelimb (p = 7.89 × 10−3), and hindlimb (p = 2.55 × 10−2) are important traits in distinguishing Johnstonei A from Johnstonei B; and head length (p = 3.30 × 10−4) and ear to eye distance (p = 2.73 × 10−2) for distinguishing Triacantha A from Triacantha B (Fig. 4). The significant non-normal variables within C. triacantha were confirmed with significant non-parametric test (Table S5).

Figure 4 Boxplots of significantly different traits between mtDNA lineages for C. johnstonei and for C. triacantha.

JA, Johnstonei A; JB, Johnstonei B; TA, Triacantha A and TB, Triacantha B.

The analysis of meristic data was based on three relevant characters (number of ear lobules, lamellae number under the 3rd finger and under the 4th toe) due to little or no variation in the other traits. Each of the three analysed characters was significantly different between Johnstonei A and B, but only ear lobule number showed a significant difference between Triacantha A and B (Fig. 4, Table S6).

The prediction capacity of significant morphological data was investigated with a linear discriminant analysis (LDA) and a Random Forest analysis (RF). Jackknife results provided 85.87% accuracy for differentiating C. johnstonei lineages based on log-transformed morphological measurements (to include SVL as a variable) and 72.34% for C. triacantha lineages. While the accuracy estimated with a RF analysis was 81.52% for C. johnstonei and 68.09% for C. triacantha.

The summary of each measured trait can be found in Table S7.

Taxonomic assessment and species description

Considering the congruence across multiple genetic delimitation methods and of these with significant morphological divergence among lineages, we provide sufficient evidence for four species, two species within Carlia johnstonei and two species within C. triacantha. Within C. johnstonei, Johnstonei A is the nominal C. johnstonei species based on a holotype from the Mitchell Plateau, a region in which extensive sampling has shown that only Johnstonei A occurs. For C. triacantha the holotype specimen is from Adelaide River, Northern Territory, a site close (∼15 km) to Triacantha A samples from Litchfield National Park (NTM R22162)—hence we suggest that Triacantha A should retain the species name. Accordingly, we here describe two new species—Johnstonei B as Carlia insularis sp. nov. and Triacantha B as Carlia isostriacantha sp. nov. In the following we provide diagnoses for the four species. Simple genetic diagnostics (mtDNA diagnostic SNPs; Table 2) are robust. For morphology alone, single traits mostly have overlapping ranges, but in combination with each other and geography, should be practical in the field.

Table 2 ND4 mtDNA diagnostic SNPs for each lineage.

The position of each SNP is aligned with Scincella vandenburghi mitochondrial genome (Park et al., 2016). For each nucleotide position is also presented the correspondent amino acid substitution. Grey background refers to which species the SNP is diagnostic for.

	10851	10864	10992	11115	11218	11365	11413	
C. johnstonei	T	Ser	A	Tyr	A	Thr	A	Met	C	Thr	A	Asn	Ta	Ilea	
C. insularis sp. nov.	A	Thr	G	Cys	A	Thr	A	Met	T	Ile	G	Ser	T	Ile	
C. triacantha	T	Ser	A	Tyr	G	Ala	C	Leu	C	Thr	G	Ser	C	Thr	
C. isostriacantha sp. nov.	T	Ser	A	Tyr	A	Thr	A	Met	C	Thr	G	Ser	T	Ile	
Notes.

a Substitution is not diagnostic for a few individuals.

Carlia johnstonei Storr, 1974 Records of the Western Australian Museum, Vol. 3, 151-165

Rough brown rainbow-skink

Holotype. WAM R43170, from Mitchell Plateau, Western Australia, in −14.866667 125.833333.

Diagnosis. Dark blackish Carlia morphologically distinguishable from geographically overlapping species with a combination of mid-dorsal scales bicarinate (two keels), more numerous supraciliares (usually 7 vs. 6—C. amax, C. munda, C. rufilatus, C. isostriacantha sp. nov., or 5—C. gracilis), larger ear aperture with numerous sharply pointed lobules (mean of 10 lobules), but typically less than in C. insularis sp. nov. (mean of 13 lobules). Further distinguished from the latter by smaller body size (mean 36.39 mm vs. 41.83 mm), reduced head depth (mean 3.59 mm vs. 4.48 mm), shorter limbs (forelimbs 9.51 mm vs. 11.45 and hindlimbs 14.82 mm vs. 17.77 mm) and less lamellae under longest finger (mean 16.75 mm vs. 19.69 mm) and toe (mean 22.83 mm vs. 26.31 mm).

Description. Snout-vent length (mm): 21.84–43.49 (N = 66, mean 36.39). Tail: 27.1–61.28 (N = 26, mean 46.04). Most specimens with separated prefrontal scales (93%) by an average of 0.32 mm (N = 50, 0.05–0.64). Ear aperture smaller (N = 62, mean 1.01, 0.50–1.44), than palpebral disc (N = 62, mean 1.31, 1.05–1.59), with many small lobules (mean 10, 5–16). Lamellae under third finger 9–20 (N = 63 mean 16.75), fourth toe 15–27 (N = 63 mean 22.83) (Table S7). Most specimens are dorsally dark brown and ventrally yellow but with either a bright or dark blue gular.

Distribution. Distributed across the sub-humid area in the Kimberley, from the northeast Berkeley River region, to the southwest King Leopold Ranges (Fig. 1). Present in humid islands in the Kimberley, mostly the northern islands and those closer to the mainland. In drier environments, this species tends to be more restricted to mesic microhabitats in rocky gorges (Russell Barrett, pers. comm., 1993–2016).

Remarks. The previous described paratype from East Montalivet Island (WAM R41462) in Storr (1974) by geographic location should belong to C. insularis sp. nov.

Carlia insularis sp. nov. (Figs. S7A, S8A, S8C and S9A) urn:lsid:zoobank.org:act: F058DFD2-799C-4242-8926-9F59AEC6FD44.

Kimberley islands rainbow-skink

Holotype. WAM R158646, from North Maret island, Western Australia, in −14.3983 124.97750. Specimen collected in 2004 by Richard How (Fig. S7A).

Paratypes. Fenelon Island: WAM R117708, WAM R117709, WAM R117710; Corneille Island: WAM R117967; West Montalivet Island: WAM R158562, WAM R158571; Don Island: WAM R158610; North Maret Island: WAM R158647 (Table S1, Figs. S8A, S8C).

Etymology. Insularis is derived from the Latin word insular, for island, since this species is restricted to islands.

Diagnosis. Morphologically similar to C. johnstonei and distinguished from this species by the presence of mid-dorsal body scales with a mix of two or three keels (Fig. 5), whereas C. johnstonei always has two keels. As mentioned previously, it is also distinguished from C. johnstonei by longer body size, higher relative head depth, longer relative limb length, more sharp lobules in the ear aperture (mean values of 13 vs. 10; Fig. 5) and more lamellae under longest finger and toe (average 3 more). Prefrontal scales are either narrowly separated or in contact, while C. johnstonei often has more widely separated prefrontals. From a genetic perspective, four sites that change amino acids in the mtDNA ND4 sequence reliably distinguish Carlia johnstonei and Carlia insularis sp. nov. (Table 2). Geographically distinct from C. johnstonei in some of the most outer islands of the Bonaparte Archipelago (see below).

Figure 5 Relevant diagnostic traits.

Irregular keeling in dorsal body scales for C. insularis sp. nov. (A) and difference in ear lobules of C. triacantha (left) and C. isostriacantha sp. nov. (right). Illustrations by Erin Walsh.

Comparison with congeners. Distinguished from remaining Australian Carlia species by a reduced upper preocular and well separated from posterior margin of second loreal scale (Hoskin & Couper, 2012); a distinct interparietal, usually seven supraciliaries, prefrontals usually separated; at least 34 mid-body scale rows, that are dorsally 6-sided, each scale with an angular free edge and strongly bicarinate, with the keels aligned to form continuous longitudinal lines; ear-opening surrounded by many small and pointed lobules (Cogger, 2014). It is endemic to Kimberley islands where C. johnstonei and C. isostriacantha sp. nov. also occur at a regional scale. See diagnosis to distinguish from C. johnstonei; and distinguishable from C. isostriacantha sp. nov. by the presence of two keeled-scales and usually seven supraciliaries instead of six.

Description of holotype. Individual with 42.01 mm as SVL, tail 69.33 mm, axilla-groin length 19.71 mm, head length 8.86 mm, head width 6.29 mm, head depth 3.85 mm, forelimb 12.47 and hindlimb 17.42 mm. Body with keeled dorsal scales, mostly two keels but some scales with three. Six supraciliares, seven supralabials, six infralabials, 19 subdigital lamellae in 3rd finger, 26 subdigital lamellae in 4th toe. Circular ear not smaller (1.37 mm) than palpebral disc (1.19 mm) with 12 sharp ear lobules. Prefrontals narrowly separately and nasals widely spaced (2.56 mm). Midbody scale rows 37, 43 vertebral scales and 62 ventral scales.

Description. Snout-vent length (mm): 27.93–51.44 (N = 35, mean 41.83). Tail: 29.05–69.98 (N = 18, mean 51.02). Most specimens with separated prefrontal scales (62%) by an average of 0.18 mm (N = 21, 0.02–0.54). Ear aperture smaller (N = 32, mean 1.27, 0.85–2.16), than palpebral disc (N = 32, mean 1.44, 1.04–2.13), with many small lobules (up to 18). Lamelae under third finger 17–23 (N = 35 mean 19.69), fourth toe 21–30 (N = 35 mean 26.31) (Table S7). Laterally and dorsally blackish brown while ventrally yellowish with sometimes a bright blue or a dark blue gular (Fig. S8C), where in breeding males (Fig. S9A) lateral midbody has a light brown almost orange colour.

Distribution. Across the northwest and outer islands of the Bonaparte Archipelago (northern Kimberley islands in Western Australia) with confirmed occurrence on the Fenelon, Corneille, East Montalivet, West Montalivet, Don, Berthier, North Maret and South Maret islands.

Remarks. Despite extensive sampling, there are no records of C. insularis sp. nov. and C. johnstonei occurring on the same islands. All islands where the former species is confirmed are either laterite or volcanic islands, whereas C. johnstonei also occurs in sandstone islands (How et al., 2006). The individuals of C. insularis sp. nov. were collected in vine thicket and deciduous vine forest habitats (Richard How, pers. comm., 2016). Despite Descartes island being relatively close to Fenelon and Corneille islands, only C. johnstonei was confirmed on this island.

Carlia triacantha Mitchell, 1953, Records of the South Australian Museum, Vol. 11, 75–90.

Desert rainbow-skink

Holotype. SAM R2697, from Adelaide River, Northern Territory, in −13.183 131.1.

Diagnosis. Species morphologically distinguished from congeners by having three strong keels in scales, prefrontals more often in contact or very narrowly separated and usually six supraciliaries. Although more work is still needed to find unambiguously diagnostic traits between this species and C. isostricantha sp. nov., C. triacantha are mostly smaller (mean 36.55 mm vs. 40.07 mm), with shorter relative head length (mean 7.24 mm vs. 8.25 mm) and fewer ear lobules (usually 6 vs. 9, Fig. 5B). Geographically diagnosis from C. isostricantha sp. nov., possible in the centre of Australia, particularly Pilbara and Macdonald ranges region.

Description. Snout-vent length (mm): 23.78–44.98 (N = 35, mean 36.55). Tail: 38.48–75.90 (N = 17, mean 60.80). Prefrontal in contact (63%) while the rest with separated prefrontals (N = 11) by an average of 0.26 mm (0.03–1.81). Ear aperture smaller (N = 30, mean 1.13, 0.64–1.73), than palpebral disc (N = 30, mean 1.41, 0.99–1.71), with often one larger anterior lobule and several small (up to 7). Lamellae under third finger 16–22 (N = 30 mean 18.83), fourth toe 23–28 (N = 29 mean 24.83) (Table S7). Dorsally brown and ventrally yellow blueish with sometimes whitish line under eye.

Distribution. Widely distributed from Pilbara in Western Australia to Northern Territory (Fig. 1). However, more sampling and genetic analyses are needed to investigate whether this species is continuously distributed from the mesic Top End to arid central Australia or if the central Top End is only occupied by C. isostriacantha sp. nov.

Carlia isostriacantha sp. nov. (Figs. S7B, S8B, S8D and S9B) urn:lsid:zoobank.org:act:EB2E9D69-8E1F-466D-8441-E2E4DD59F96E.

Monsoonal three-keeled rainbow-skink

Holotype. WAM R171420, from Prince Regent Nature Reserve, Western Australia, in −15.98972 125.32944. Specimen collected in 2010 by Paul Doughty (Fig. S7B).

Paratypes. WAM R168173 (Boongaree Island), WAM R168675 (Katers Island), WAM R171211 (Darcy Island), WAM R171905 (Wargul Wargul Island), WAM R171906 (Molema Island), WAM R171908 (Sunday Island), WAM R171909 (Balami ridge), WAM R171916 (Lachlan Island), WAM R171921 (Storr Island), WAM R171933 (Balami ridge) (Table S1, Figs. S8B, S8D).

Etymology. Isostriacantha is derived from equal in greek (isos) with triacantha, (three spines, referring to the three keels in scales) due to the difficulty of morphologically distinguishing from its sister species C. triacantha.

Diagnosis. As similar to C. triacantha, this species is morphologically distinguished from other Carlia species by having three strong keels in scales, prefrontals more often in contact or very narrowly separated and usually six supraciliaries. As above-mentioned, in contrast with it closest relative, C. triacantha, this species has longer body size, a relatively longer head and tends to have more ear lobules, on average nine very small lobules (Figs. 4 and 5, Table S7). Another possible trait to distinguish between these species is a white line that begins posterior to each hind limb and can extend to midway through the tail (Fig. S9B). This trait is more evident in freshly caught individuals, or photographs of them, than in long preserved specimens and needs to be further tested through more observations on genetically typed individuals. Genetically diagnosed from C. triacantha, by three ND4 mtDNA sites (Table 2) and geographically by occurring in the Kimberley, although geographic diagnoses in Northern Territory requires further work.

Comparison with congeners. This species can be separated from most Australian Carlia species by an upper preocular reduced and well separated from posterior margin of second loreal scale (Hoskin & Couper, 2012); a distinct interparietal, with usually six supraciliaries, prefrontals usually in contact or narrowly separated; 28-36 rows of mid-body scales, that are dorsally 6-sided triscupid, each usually with an angular free edge and strongly keeled; often one larger anterior lobule with many small lobules in a round ear-opening that is smaller than palpebral disc, while the palpebral disc occupies much more than half of lower eyelid (Cogger, 2014). Specifically with potentially sympatric species, C. johnstonei, C. amax, C. rufilatus, C. gracilis and C. munda, this species can be identified by the presence of three strong keels in scales, prefrontals usually in contact, six supraciliaries and absence of white lateral line anterior to the forelimbs. To distinguish from its sister species, C. triacantha, see Diagnosis above.

Description of holotype. Male individual with 43.22 mm as SVL, tail 63 mm, axilla-groin length 20.06 mm, head length 8.79 mm, head width 6.57 mm, head depth 3.83 mm, forelimb 13.06 mm and hindlimb 19.34 mm. Body with three keeled dorsal scales. Six supraciliares, seven supralabials, six infralabials, 17 subdigital lamellae in third finger, 23 subdigital lamellae in fourth toe. Horizontal ear wider (1.83 mm) than palpebral disc (1.57 mm) with 13 small sharp ear lobules (one anterior larger). Prefrontals in contact and nasals widely spaced (2.34 mm). Midbody scale rows 35, 38 vertebral scales and 52 ventral scales.

Description. Snout-vent length (mm): 24.72–49.12 (N = 68, mean 40.07). Tail: 29.1–86.68 (N = 39, mean 61.55). Prefrontal in contact (73%) while the rest with separated prefrontals (N = 19) by an average of 0.12 mm (0.01–0.37). Ear aperture smaller (N = 67, mean 1.33, 0.69–1.96), than palpebral disc (N = 67, mean 1.46, 0.92–1.81), with often one larger anterior lobule, many small (up to 13) and sometimes one superior. Lamelae under third finger 11–24 (N = 62 mean 19.27), fourth toe 18–30 (N = 62 mean 24.82) (Table S7). Dorsally brown and ventrally yellow blueish, with a light line under eye to ear, and often with a very light whitish line in the back of hindlimbs to tail if not regrown (Fig. S9B).

Distribution. Widespread across the Kimberley and adjacent (mostly southern Kimberley) islands in Western Australia, with isolated records in the western Gulf region, spanning the border of the Northern Territory and Queenland (Fig. 1).

Remarks. Afonso Silva et al. (2017) found one genetically discordant sample with mtDNA of C. isostriacantha sp. nov. and nuclear of C. triacantha from the Victoria River region (ABTC61613, Table S1). This suggests a need for further regional surveys and genetic studies, particularly in the Northern Territory where only a few specimens with tissues were detected, to define the boundaries of both species, at geographical and morphological level.

Discussion

We used extensive genetic and morphological data to identify two new species of Rainbow skinks, Carlia insularis sp. nov. (Johnstonei B lineage) and Carlia isostriacantha sp. nov., (Triacantha B lineage), in an understudied region of Australia, the Kimberley. We also redefined diagnoses and geographic distributions of Carlia johnstonei and C. triacantha. Our work takes advantage of recent progress in techniques for obtaining large-scale sequence data and in methods for species delimitation, as part of a broader integrative taxonomic approach. These advances are particularly important for identifying cryptic species, such as those described here, where morphological evidence alone is often insufficient for reliable species identification.

Evidence for new cryptic species

A previous phylogeographic study with >2,000 loci Afonso Silva et al. (2017) revealed two new candidates species in the Carlia genus. The current work confirms these are new species using three robust hypothesis-driven validation methods based on several independent sets of genes from the larger exon dataset. The use of multiple different methods provides a robust test for the previous discovery in Afonso Silva et al. (2017), and further validates the proposed species delimitation.

Although the existence of C. insularis sp. nov. and C. isostriacantha sp. nov. is well supported in the genetic data, distinguishing these species morphologically is more difficult due to their cryptic nature. The genus Carlia generally has few diagnostic taxonomic characters that allow for the separation of species using morphology. Even for C. johnstonei and C. triacantha as currently recognised, there are only a few morphological characters that effectively distinguish between these species, mainly the number of keels on the dorsal scales and the arrangement of ear lobules (Storr, 1974). However, morphological measurements broadly overlap between both C. johnstonei and C. triacantha lineages. Despite these issues, we were able to find statistically significant differences in morphology across both measurements and meristic data, supporting the presence of these lineages as different species.

Differences in body size, head and limbs traits as well as ear lobule numbers help in distinguishing the lineages. Morphological variation across each pair of taxa is strongly affected by body size (SVL), with the newly described species being larger than their respective sister taxa. The same is observed for the other significant traits, even after accounting for size. Although for both species, there are some overlap between morphological groups, there was more morphological similarity between the C. triacantha lineages than between C. johnstonei lineages (Fig. 3), likely reflecting the shallower divergence seen within C. triacantha.

Though we were able to identify a few distinct morphological traits, using morphology alone to identify individuals will remain a challenge without a reference to geography. For the two lineages within C. triacantha, even geography is a poor guide for the central Northern Territory region. Therefore, for more reliable diagnosis, we follow the suggestion of Renner (2016) and include a set of diagnostic mtDNA SNPs to distinguish between C. johnstonei and C. insularis sp. nov., and between C. triacantha and C. isostriacantha sp. nov. These SNPs can be easily assessed by cheaper Sanger sequencing of the mtDNA gene ND4 (primers and protocol in Afonso Silva et al., 2017).

Biodiversity significance of the two new species

C. insularis sp. nov. is an important addition to the known biodiversity of the Kimberley islands. This region has recently been the focus of several studies that have documented unique biodiversity communities, namely in terms of vegetation (Lyons et al., 2012), avifauna (Pearson & Caton, 2013) and herpetofauna (Doughty et al., 2012; Palmer et al., 2013). Studies to understand the biodiversity value in this region are also of importance to conservation, as this area is being considered as a biodiversity refuge for fauna vulnerable to the invasive Cane Toad (Palmer et al., 2013). Although the west Kimberley region has several endemic species, only a few are endemic just to the islands, namely a blindsnake (Ellis, 2016) and several land snails (Criscione & Köhler, 2013; Criscione & Köhler, 2014), making the discovery of C. insularis sp. nov. very significant. But more island-endemics reptiles are expected to be described, since Palmer et al. (2013) suggested the occurrence of a few potential new species (including samples that correspond to C. insularis sp. nov.) that have not yet been described.

Although our genetic data allows us to describe C. isostriacantha sp. nov. as a new species, further collecting and analyses are needed across central Northern Territory for this and other taxa (also suggested in Rosauer et al., 2016). Specifically, there is a need to identify the geographic distributions of C. triacantha sensu stricto and C. isostriacantha sp. nov., as well as to examine morphological divergence in this poorly sampled region. In a group of Ctenotus skinks, Rabosky et al. (2014) highlight how intraspecific morphological variability and geographic sampling gaps caused an inadequate understanding of biological diversity. As with Ctenotus, we suspect that many other species in the Carlia genus may yet require taxonomic revision. Potter et al. (2016) have also suggested unknown lineage diversity in another Carlia species in the Australian Monsoonal Tropics, which may lead to the description of additional Carlia species, particularly on the islands off the northeast Top End.

Advantages and issues of using MSC methods

A key element of our analysis was the use of multispecies coalescent (MSC) methods, including pioneering the application of StarBEAST2 to Bayes Factor species delimitation (BFD). MSC models are a robust approach that better describes species formation by considering coalescent processes; however, methods based on the MSC are typically computationally intensive. To surpass this limitation, we subsetted independent smaller sets of loci from around 2,300 loci, which also has the advantage of producing multiple replicate results that may be compared to confirm that estimated parameter values are robust to the choice of loci.

BFD using SNAPP and StarBEAST2 requires sampling from different power posteriors, including sampling purely from the prior. We found that convergence was difficult to achieve for our data set when BFD StarBEAST2 was used to sample from the prior with more than 20 loci. Despite this limitation, BFD StarBEAST2 has advantages over existing methods for species delimitation. Compared to SNAPP which requires unlinked SNPs, StarBEAST2 can extract much more information from each locus. Compared to BPP, StarBEAST2 has many more options for substitution models, population size models, and relaxed clock models.

Conclusions

As Oliver, Keogh & Moritz (2015) express, most genetically divergent lineages within species remain invisible to other scientific work, like conservation assessments and management planning. This reinforces the need to evaluate whether genetically distinct lineages within species should be formally described. Here we validate and describe two new species of rainbow skinks in the northwest of Australia, a highly biodiverse region of Australia that is still relatively understudied. Using an integrative taxonomic approach, we employ three MSC methods, including the application of a new approach to delimit species, as well as integrating morphological data to provide strong evidence for these two new species. This work brings the number of Australian Carlia to 26 species. However, further such work is needed across the Australian Monsoonal Tropics, since deeply divergent lineages within species of lizards are the norm in this region.

Supplemental Information

Table S1 Tissue and specimens list

Tissue and specimens list with museum origin, mtDNA lineage, for which analysis samples were used (SD, species delimitation, M, morphology), sex information if available and location. ∗ Correspond to the genetically discordant sample with mtDNA of Triacantha B but nuclear of Triacantha A (with no evidence of admixture in Afonso Silva et al. (2017)).

Click here for additional data file.

Table S2 jModelTest substitution models

jModelTest substitution models used with the two StarBeast2 datasets and fragment length of loci. Loci designation based on Anolis carolinensis genome and sequence size that was retrieved for all used samples for each locus.

Click here for additional data file.

Table S3 Priors used for Starbeast2 and SNAPP analyses

Click here for additional data file.

Table S4 Summary of MANOVA results testing for significant interaction with mtDNA lineage within C. johnstonei

The results for testing normality and heteroscedasticity are also presented for both the log-transformed and the log and size-corrected dataset. Bold are significant p-values for the MANOVA results. After removing samples with missing data, analyses were performed with a total of 92 specimens.

Click here for additional data file.

Table S5 Summary of MANOVA results testing for significant interaction with mtDNA lineage within C. triacantha

The results for testing normality and heteroscedasticity are also presented for both the log-transformed and the log and size-corrected dataset. Bold are significant p-values for the MANOVA results. ∗, non-normal variables with significant support with the non-parametric Wilcoxon test. After removing samples with missing data, analyses were performed with a total of 94 specimens.

Click here for additional data file.

Table S6 Summary of Generalized Linear modelling with a Poisson distribution analyses for relevant meristic variables

Summary of Generalized Linear modelling with a Poisson distribution analyses for relevant meristic variables, presenting estimates and respective confidence intervals (C.I.). Bold correspond to significant p-values. After removing samples with missing data, analyses were performed with a total of 85 and 83 specimens for C. johnstonei and C. triacantha, respectively.

Click here for additional data file.

Table S7 Descriptive table with measurements and meristic data for each main lineage within C. johnstonei and C. triacantha

Click here for additional data file.

Figure S1 mtDNA ND4 maximum likelihood phylogenetic tree of Carlia triacantha and Carlia johnstonei

mtDNA ND4 maximum likelihood phylogenetic tree of Carlia triacantha and Carlia johnstonei, from Afonso Silva et al. (2017), with specimens that were analysed by morphological analyses. Sample label includes tissue number, original ID and sampling location.

Click here for additional data file.

Figure S2 Stepping-stone computation of marginal likelihoods for Bayes factor species delimitation

This method calculates the marginal likelihoods from the area under the likelihood posterior curve. Two replicate chains were run for each method and dataset. The mean likelihoods are plotted with + symbols for one chain, and × symbols for the other. Segmented lines approximating the curve are plotted by connecting the likelihoods averaged for both chains.

Click here for additional data file.

Figure S3 Measurements taken from photos

(A) Lateral view of specimen with ear aperture length (EAL), eye to ear distance (EED) and palpebral disc length (PDL). (C) Nasal separation (NS) measured in dorsal view. (C) and (D) correspond to forelimb (FLL) and hindlimb length (HLL) measurements in ventral view. Photos by Damien Esquerré.

Click here for additional data file.

Figure S4 Box plots for all log transformed variables

JA, Johnstonei A; JB, Johnstonei B; TA, Triacantha A; TB, Triacantha B.

Click here for additional data file.

Figure S5 Box plots for log and size corrected variables

JA, Johnstonei A; JB, Johnstonei B; TA, Triacantha A; TB, Triacantha B.

Click here for additional data file.

Figure S6 PCA loadings and variables importance of PCA with log transformed data and with size corrected data for both species

PCA loadings and variables importance of PCA with log transformed data (A, B) and with size corrected data (C, D) for C. johnstonei and C. triacantha.

Click here for additional data file.

Figure S7 Dorsal and ventral view of holotypes

Dorsal and ventral view of holotypes of C. insularis sp. nov. (A), specimen WAM R158646, and C. isostriacantha sp. nov. (B), specimen WAM R171420. All photos by Damien Esquerré.

Click here for additional data file.

Figure S8 Dorsal and ventral paratype photos

Dorsal and ventral paratype photos for C. insularis sp. nov (A, C) and for C. isostriacantha sp. nov. (B, D). All photos by Damien Esquerré.

Click here for additional data file.

Figure S9 Photos with live animals

Photos with live animals showing breeding colours of C. insularis sp. nov. (A, photo by Russell Barrett) and a potential diagnostic trait in C. isostriacantha sp. nov. (B, photo by Mark Allen). The white arrow points to the potential white line trait that distinguish this species from C. triacantha.

Click here for additional data file.

We thank the Museum and Art Gallery of the Northern Territory, Museum Victoria, South Australia Museum and Western Australia Museum for specimens, in particular to Ryan Ellis for his assistance. We are also grateful to members of the Moritz lab for ongoing discussions and comments relating to this study, and in particular to Mozes Blom, Joshua Peñalba, Rebecca Laver, Sally Potter, as well to Emma Sherratt, Liam Bailey and Thomas Merkling for advice on analyses and comments on the manuscript. We are also grateful to Fabricius Domingos and an anonymous reviewer for helpful comments. We acknowledge credit and thank to Erin Walsh, Damien Esquerré, Russell Barrett and Mark Allen for illustrations and photos.

Additional Information and Declarations

Competing Interests

Author Contributions

Animal Ethics

Data Availability

New Species Registration

The authors declare there are no competing interests.

Ana C. Afonso Silva conceived and designed the experiments, performed the experiments, analyzed the data, wrote the paper, prepared figures and/or tables, reviewed drafts of the paper.

Natali Santos performed the experiments, analyzed the data.

Huw A. Ogilvie conceived and designed the experiments, performed the experiments, analyzed the data, contributed reagents/materials/analysis tools, reviewed drafts of the paper.

Craig Moritz conceived and designed the experiments, contributed reagents/materials/analysis tools, wrote the paper, reviewed drafts of the paper.

The following information was supplied relating to ethical approvals (i.e., approving body and any reference numbers):

The Australian National University Animal ethics office provided full approval for this research permit with the number A2012/14.

The following information was supplied regarding data availability:

Silva, Ana (2017): Data and code to describe Carlia johnstonei and C. triacantha. figshare. https://doi.org/10.6084/m9.figshare.4621963.v3.

The following information was supplied regarding the registration of a newly described species:

Publication LSID: urn:lsid:zoobank.org:pub:A7B29F16-079F-48BA-B4BE-3EC9A3D80D34.

Carlia insularis: LSID: urn:lsid:zoobank.org:act:F058DFD2-799C-4242-8926-9F59AEC6FD44.

Carlia isostriacantha: LSID: urn:lsid:zoobank.org:act:EB2E9D69-8E1F-466D-8441-E2E4DD59F96E.

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
