# Peer review of "Validation and description of two new north-western Australian Rainbow skinks with multispecies coalescent methods and morphology"

_PeerJ, doi:10.7717/peerj.3724_

## Round 0.1 · original submission · Minor Revisions

Overall, I enjoyed reading this manuscript and found it highly interesting. While I agree with reviewer 1 that there is overlap between this manuscript and the ME manuscript I feel the description of species in this manuscript gives it sufficient merit as stand alone. Having said this there is still work to do to make this manuscript acceptable for publication. Both reviewers have done a fantastic job highlighting potential improvements to the manuscript. In particular, two areas I think that would benefit the manuscript are improving the quality of figures and improving the methodology section. Both reviewers provide in depth details for this, so please take care in making some of their suggested changes. Once you have made changes I look forward to receiving your revised manuscript.

Reviewer 1 ·

Basic reporting

The authors present a standard species delimitation paper using all appropriate data types - combining both next-generation sequencing technologies with traditional morphological approaches. My main concern is that this paper is heavily reliant on an unaccepted manuscript and that I needed to thoroughly read through this manuscript to make sense of their experimental design and description of their study system. If this manuscript was accepted, this would not be an issue - however, the PeerJ submission cannot stand alone. In my opinion, the first manuscript must be accepted for this work to be accepted. The authors provided their submission to Molecular Ecology. While I have some questions/concerns about that manuscript, I leave it to the ME reviewers to assess their work. If accepted, the authors do need to expand on their exon capture design in the PeerJ manuscript.

For the most part, manuscript was clear and well-written.

The formatting of the citations needs work: missing italics for scientific names, year placement, inconsistent capitalization of title names, inconsistent capitalization of journal names.

All tables + figures lack names and captions. Figure 1 is difficult to read - please make sampling localities larger and it's hard to see the color with the black border around symbols. It's impossible to read the distance scale. Support on tree is tiny - perhaps replace 100% bootstrap support with symbols?

Experimental design

The description of BPP appears to be in conflict with itself. The authors state they use the joint estimation of species tree inference and species delimitation, but also state that they used a guide tree. Species tree delimitation in BPP should be done without a guide tree. The parameters for BPP are explained but these are lacking for both *BEAST and SNAPP - especially since SNAPP is a relatively new method, there is a constant stream of new revelations about the influence of priors, and the SNAPP results did a poor job of supporting their expected topology. Furthermore, it does not appear that the authors conducted more than one independent run of SNAPP per dataset. More runs should be conducted to assess convergence. For the *BEAST tree, the authors chose to analyze 20 randomly? selected genes out of >2,000. They only replicated this analysis twice. I see this as a missed opportunity but not required. Finally, all of these methods require a priori assignment of individuals to species/tips. I cannot find a discussion of this and what logic was used for these definitions. I assume it stems from the ME manuscript. However, I am not fully convinced that lumping Johnstonei A into a single group is appropriate, as there is quite a bit of structure within it. The authors should do a better job at explaining their reasoning and the support for these a priori definitions.

Validity of the findings

As stated above, there is much reliance on an unpublished manuscript. The statements made under the heading 'Advantages and issues of using MSC methods' seem out of place and irrelevant to me. Also, some of the comments within this section are observations of results not stated in the results section.

·

Basic reporting

The manuscript uses clear and unambiguous English, and I only comment on a few issues below. Specific hypotheses are stated and satisfactorily tested.

Experimental design

The research questions are clear, and the used methods are robust. Particularly, I urge the authors to make their morphological dataset available (as per PeerJ standards). I also point to a few concerns regarding the morphological analyses below.

Validity of the findings

The conclusions are well stated and discussed. I commend the authors for the description of the two new species, which I consider a great feature of the manuscript (since many species delimitation papers do not actually describe species).

Additional comments

Please find my complete line-by-line review below. I also raise five points that I consider of particular relevance when preparing a revised version of the manuscript.

1. Clarify how each dataset was used (comments on lines 136–137 below).
2. Provide a better description of the molecular dataset (comments on lines 136–137 below).
3. The use of ambiguous sites (unphased data) in BPP and IMgc (comments on lines 165–166 below).
4. Concerns on outliers, missing data, morphological analyses, and availability of the data (comments on lines 236–237 below).

Title: I like the title, and it satisfactorily describes the manuscript. However, since two new species are described (which is a strong component of the manuscript, not always present in species delimitation papers), the authors might want to use ‘description’ instead of ‘delimitation’ (or slightly changing the title to make this information clearer)? This is not a requirement, but only a suggestion for the authors to consider.

Lines 56–57: Although I do not disagree with your cryptic species definition, this issue have been somewhat discussed in the literature and I would suggest you to please use the definition of Bickford et al. (2007). Otherwise, if you have strong reasons to keep your current definition, please provide a reference for it.

Line 66: Please consider changing ‘presence’ to ‘species limits’ or ‘species boundaries’, or even ‘taxonomic validity’.

Line 72–75: As far as I am aware, the first authors to use the terms ‘discovery’ and ‘validation’ when referring to these different approaches to species delimitation were Ence and Carstens (2011) (and not O’Meara 2010). I might be mistaken, but please double check it and correct the citation if appropriate.

Line 84: The citation should read ‘Carstens’ and not ‘Carsten’.

Line 91: Please consider adding ‘by alternative sources of data’ (or similar) after ‘somehow corroborated’.

Line 94: Since this is the first appearance of the species names, please provide the complete names (i.e., with author and year).

Line 96: Could you please provide a short sentence detailing where the Kimberley is? Although this information is somehow present in the map, I believe this information would improve understanding the biogeographic patterns behind the species distribution (since many readers might not be familiar with Australian biogeography).

Line 105: Please detail to whom the unpublished data belongs to (e.g., Silva & Moritz unpublished data).

Line 113–122: I feel like this section should be written in past tense (e.g., ‘reanalysed’ instead of ‘reanalyse’ and so on). Please change it accordingly if you believe this is appropriate.

Line 117: The current phrasing seems to (wrongly) imply that Leaché et al. (2014) are the authors of SNAPP. I would suggest the authors to use the terminology proposed by Leaché et al. (2014) – BFD*, (e.g., BFD* (Leaché et al 2014) using SNAPP (Bryant et al. 2012)), and correct it for consistency throughout manuscript.

Line 118: Maybe the authors would like to stress the use of BFD with StarBEAST2, like they did in the abstract (‘We introduce an approach that is based on the recently published method StarBEAST2’).

In relation to my last two comments, the authors might want to consider using ‘BFD* SNAPP’ and ‘BFD* StarBEAST2’ throughout the manuscript, so that the distinction between using these programs for species delimitation and merely citing the programs is clearer.

Line 119: Please change ‘potential hypotheses’ to ‘potential species hypotheses’.

Lines 136–137: The sentence ‘all analyses were performed with five individuals for each of 
the four identified lineages’ certainly refers to the exon capture data, which was used in the species delimitation analyses. However, since the mtDNA information + morphology data were used in other analyses, please rewrite for clarity.

In the same reasoning, I urge the authors to please use this first part of the Material and Methods to clearly state how and why each dataset was used, in order to relate it to the other sections of the MM (i.e., exon capture (20 individuals, ??? localities) used for coalescent species delimitation analyses; mtDNA and morphology (200 individuals, ??? localities) used to seek for morphological differences/ diagnostic characters among the lineages).

Also, please provide a better description of the molecular dataset in this section (e.g., the number of used loci only appears much later at line 197) and how it was collected. Although this is detailed in Silva et al (in review), it would certainly improve readability and make it easier for the readers to understand and assess the molecular methods. I am not suggesting repeating the methods of Silva et al (in review) but, actually, only providing a summary of it – for example, of Silva et al (in review) lines 150–151 (subset of samples), lines 181–189 (alignment and quality check), and lines 295–302 (number of loci, length, etc).

Line 137: It is unclear what you mean by ‘the same C. amax samples’. The same used by Silva et al. (in review)? Please rephrase for clarity.

Line 145–146: Maybe using ‘investigate species limits’ or ‘boundaries’ would be a better description than ‘investigate species delimitation’?

Lines 147–148: please change ‘and’ to ‘using’ (multilocus and SNP data).

Lines 149–150: May you please provide a justification of why you believe randomly selecting different loci subsets (i.e., disregarding information content) is a robust strategy?

Lines 150–151: Please provide a reference for the MSC assumptions.

Lies 154–156: Thank you for evaluating within-loci recombination. However, please provide a justification of why you only applied this procedure to one exon set and not both.

Line 155: Please cite the appropriate IMgc reference (Woerner, A.E., M.P. Cox and M.F. Hammer. (2007) Recombination-Filtered Genomic Datasets by Information Maximization. Bioinformatics 23:1851-1853).

Line 161–163: Did you try using different priors? Please provide a justification of why these priors are deemed adequate (maybe biological information on the group, previous research, etc).

Lines 165–166: As far as I can tell, the exon capture data is not phased (lines 178–180 of Silva et al (in review)). If that is indeed the case, then using all columns in the likelihood calculation is not appropriate, since the MSC assumes that you are using alleles. I believe you results will not change, but the BPP analyses need to be re-run either excluding the ambiguous sites or after phasing the data. If you decide to follow the first suggestion, may you also please provide a percentage of missing (N) and ambiguous sites in both your exon capture sets?

Also, IMgc does not allow the use of ambiguity codes, and and treats (-) as indels. How was your data transformed so the program could be used?

Line 168: It is customary to run BPP for at least 500,000 generations. I have seen BPP analyses converge much before that, in which case your 200,000 generations would be fine. May you please provide some comments on that to justify the number of generations used?

Line 179: I see that you used PartitionFinder in Silva et al (in review). Why did you use JModelTest in this manuscript instead? Since the partition scheme can potentially change branch lengths and topology, it could also potentially change the Bayes factor of your different hypotheses. Alternatively, you could use the native BEAST2 partition selection (Wu et al. 2013), which is supposedly much more appropriate when using BEAST.

Line 182: Please change ‘saved’ to ‘sampled’.

Line185: Please specify what you mean by ‘evaluating’ the trees in DensiTree.

Line 186: Please change ‘bests’ to ‘best’.

Line 199: How many replicates per SNP dataset?

Line 199: Please change ‘insure’ to ‘ensure’.

Line 201: Which ‘support values’ are you referring to? Please clarify.

Lines 219–220: Please provide a reference for how supralabials, infralabials, and supraciliars were counted, or provide more specific details so other researchers could potentially count exactly the same characters. You might also want to provide a picture showing how counting was performed, or maybe incorporate it in Figure S1.

Line 222: Please detail how keels were counted and used (e.g., from all midbody scales? If not, how were the counts determined? Was there variation within specimens? If so, did you average the counts to use in the analyses? etc).

Line 223: Please correct ‘analyzed’ to British/ Australian spelling.

Lines 227–229: Although I understand these are very time consuming meristic counts, and acknowledge the reason why they were only counted in the holotypes, some more traditional taxonomists could argue that this information should also be available for the paratypes. To overcome this issue (without having to go back to the specimens), you could maybe provide high-resolution photographs of the paratypes? The current photographs (Figure S6) would not allow someone to actually count it (if needed).

Line 232–233: Although the sentence ‘We investigated the relationship of each linear measurement with size (per mtDNA lineage)’ suggests that you separated your morphological data into the four lineages using mtDNA, this is not clearly stated in the manuscript. Since cryptic species are, supposedly, not easily distinguishable morphologically, in order to clarify this issue, and to further support the analytical framework, I suggest the authors to provide a ND4 phylogenetic tree (ML or NJ would suffice in this case) containing all specimens for which morphological data was analysed (and please see my comment on lines 136–137 above).

Line 234: From where did you ‘extracted size-corrected residuals’? Maybe from regressions between SVL and each variable? If that is so, please clarify (since you only mention ‘plotting’ the variables against each SVL).

Line 236–237: Apart from testing for normality and heterocedasticity, did you check for the presence of outliers in your dataset? If not, please test for the presence of outliers, and make sure no univariate or multivariate outliers are present (since they can bias the analyses).

Also, you did have some missing data (Table S5), probably because of damaged specimens. However, multivariate analyses cannot handle NAs, and most MANOVA implementations in R will automatically exclude the whole case (all variables of a specimen that has a single NA) or simply fail to run. The same is true for the PCA, which will probably fail unless you use na.omit. Thus, your final dataset was smaller than your original dataset (with NAs). Can you please provide details on how you approached this issue in your analyses? If you did not input your missing data, please mention how large your final dataset was (i.e., how many specimens (or degrees of freedom) were actually used).

Moreover, and in accordance to PeerJ standards, please provide a table with the raw measurements and counts of all your 200 specimens (since Table S5 only reports average values).

Line 237–238: Please provide the R version you used, and the complete reference for such version (e.g., R Core Team, 2016).

Line 241: Please mention that the command you used (prcomp) is from the ‘stats’ package.

Line 243: I believe you probably used ‘ggplot2’ (and not ‘ggplot’, discontinued in 2008). Also, please provide a reference for the ‘ggplot2’ package.

Line 246–250: When you mention that you could not account for sex, does that means you used all individuals or only one sex? And what do you mean by ‘remaining sexed individuals’?

Line 245–246: From your MM I thought you combined measurements and meristic variables in only one MANOVA (‘with a MANOVA’), but your results suggest otherwise. Please re-write for clarity.

However, why not running only one (size-corrected) MANOVA including morphometric and meristic characters?

Did your meristic variables present normality and homoscedasticity?

I am assuming your meristic characters probably lack normality and are heteroscedastic, but please report it if that is not true. In any case, the morphometric variables likely show some degree of multicollinearity, and the same should be true for the meristic variables. Taken together, these issues (assumption violations) might affect the results of your MANOVAs and LDA. Please provide an explanation of how you expect that these issues were appropriately assessed in your analyses. Only as a suggestion (not a requirement), you might also consider using an alternative approach that overcomes all assumption issues, by the use of machine learning algorithms. For example, Murphy et al. (2016) used a Guided Regularized Random Forest that can test for differences among species, determine predictor importance, and determine the variables assignment accuracy.

On another perspective, since morphologically diagnosing the new species is not particularly easy, you might also consider running a stepwise discriminant analyses (one between each sister species) to select the best diagnostic character. The same procedure can be done using the Random Forest analysis mentioned above.

Line 251: Sentence is unclear (‘statistically significant between on mtDNA lineage’). Please re-write.

Line 253: Please add “implemented in ’the package’ MASS” to the sentence.

Line 255: I highly endorse providing a molecular diagnosis for the species. Thisis great. However, Table 2 indicates that you used the mitogenome of Scincella vandenburghi to do so. May you please provide more details on this section? (e.g., reference for the Scincella vandenburghi mitogenome, alignment method, did you visually selected SNPs? Did you use all available diagnostic SNPS? etc).

Line 284: Please exclude ‘as well’ after ‘and’.

Line 292: Please exclude ‘incidentally’.

Line 296–297: Was there a better topology estimated by SNAPP? I understand the caveats about using SNAPP, but since it is the only species tree method that used your whole dataset (in this manuscript), you might consider reporting a SNAPP species tree in the supplementary material.

Line 305: Please change ‘just’ to ‘only’.

Line 307: Please change ‘explained just’ to ‘only explained’.

MANOVA results: The authors might consider reporting the value of the statistics and the degrees of freedom alongside the reported p-values, or direct the reader to Table S4.

Line 336: Please change ‘with’ to ‘within’.

Line 339: ‘clese’ = ‘close’? Please report how close these sites are (approximate distance in kilometres).

Line 353: Please name the species that have ‘6 or 5’ supraciliars.

Line 355–356: Since this is the diagnosis, please provide specific details on how different the species are – mean body size, head depth, limb length in both species; how many ‘less lamellae’?

Line 382–390: Same as above comment.

Line 431: Please exclude ‘to be done’.

Line 432–433: same as above comments on diagnosis.

Line 456–457: Please change ‘distinguishing this species morphologically’ to ‘morphologically distinguishing…’.

Line 458: Although I acknowledge the similarity between the two sister species, I don’t believe stating that ‘Morphological diagnosis as for C. triacantha’ is suitable for a diagnosis section. Please exclude this sentence and modify the diagnosis accordingly (stating the differences in length, etc, as suggested above for the other diagnosis).

Line 492: The authors might consider using this section to remark distribution differences between this species and its sister species (since it can be used, alongside morphology, to diagnose the species in the field).

Line 507: Please change ‘taxonomic’ to ‘taxonomy’.

Line 525: Please delete ‘still’ from ‘we were still’.

Line 528: Please change ‘as well ear lobule numbers help distinguish’ to ‘as well as ear lobule numbers help distinguishing’.

Line 592: Please change ‘taxonomic’ to ‘taxonomy’.

Figure 2: In my version of Figure 2, ‘C. amax’ is almost not readable in the tree. Maybe it was a problem when submitting and uploading the figure (but please double check it).

Line 746: Please change ‘datasets’ to ‘dataset’.

Line 753: I believe ‘significant’ means ‘significantly different traits’. Please change accordingly to clarify this issue.

Line 759: Please clarify that (A) illustrates the dorsal view.

Line 760: Please use ‘Carlia isostriacantha sp. nov.’.

Table 2: C. insularis sp. nov. (last period is missing).

Supplementary Material: Please provide tables/figures caption for each Supplementary file. I suggest embedding these to each file before submission.

Table S1: Please clarify in the caption what ‘Analysis’ means in the table.


References

Bickford, D., D. J. Lohman, N. S. Sodhi, P. K. Ng, R. Meier, K. Winker, K. K. Ingram, and I. Das. 2007. Cryptic species as a window on diversity and conservation. Trends in Ecology & Evolution 22: 148–55.

Ence, D. D. and B. C. Carstens. 2011. SpedeSTEM: a rapid and accurate method for species delimitation. Molecular Ecology Resources 11: 473–80.

Murphy, J. C., M. J. Jowers, R. M. Lehtinen, S. P. Charles, G. R. Colli, A. K. Peres, Jr., C. R. Hendry, and R. A. Pyron. 2016. Cryptic, Sympatric Diversity in Tegu Lizards of the Tupinambis teguixin Group (Squamata, Sauria, Teiidae) and the Description of Three New Species. PLoS One 11: e0158542.

Wu, C. H., M. A. Suchard, and A. J. Drummond. 2013. Bayesian selection of nucleotide substitution models and their site assignments. Molecular Biology and Evolution 30: 669–688.

---

## Round 0.2 · accepted · Accept

I think the authors have done a great job of addressing all of the concerns raised by the reviewers.

·

Basic reporting

The manuscript uses clear and unambiguous English, and I only provide a few suggestions in the comments section. Specific hypotheses are stated and satisfactorily tested.

Experimental design

The research questions are clear, and the used methods are robust.

Validity of the findings

The conclusions are well stated and discussed. I commend the authors for the description of the two new species, which I consider a great feature of the manuscript (since many species delimitation papers do not actually describe species).

Additional comments

I congratulate the authors for this much-improved version of the manuscript, which satisfactorily solved all potential issues present in the first version. I am particularly happy that molecular analyses were repeated using phased data and other improvements, and the methods are definitely more robust now. The same is true for the morphological analyses, which were greatly improved.

This is indeed a great species delimitation/ description paper, and I believe it is also a great contribution to the field and to our knowledge on Australian fauna.

Although I have reviewed the previous version of the manuscript, I thoroughly read the whole manuscript again. I provide a few small suggestions/ remarks below:

Line 32: Maybe change ‘does this’ by ‘tests species boundaries’ or similar statement.

Line 37: Place ‘(2163 exons)’ after dataset (and maybe exclude ‘from’).

Line 43: Please move ‘results’ and place it after ‘highly congruent’.

Lines 39–40: You already used Carlia in the lines above, and only now refers to the vernacular name (rainbow skinks). Maybe use only Carlia in the abstract, or refer to the vernacular name before?

Line 114: Delete the comma after the reference.

Line 121: It seems like exploiting ‘a large molecular dataset’, and not ‘molecular datasets’ as a general reference, would be a better way to provide such information. This would make the reference to Bragg et al 2015 a little out of place, then maybe cite Afonso Silva et al 2017 or simply delete it.

Line 154: Please change ‘do’ to ‘perform’.

Line 242: Please move ‘previously’ and place it before ‘described’.

Line 593: Maybe change ‘detected’ to ‘collected’.

Line 598: Delete comma after ‘nov.’.

Line 601: Please change ‘taxonomic’ to ‘taxonomy’.

Line 625: Maybe change ‘presence’ to ‘description’.

Line 620: Change ‘there are’ to ‘there is’.

Line 652: Please delete ‘, namely’.

Line 662: Please change ‘allows’ to ‘allowed’.

Table S7: Please specify in the table caption what exactly are the reported numbers, i.e. what we see in each line and in the brackets (Probably – Line1: mean (n), Line 2: range).